# Homologous and Heterologous Anti-COVID-19 Vaccination Does Not Induce New-Onset Formation of Autoantibodies Typically Accompanying Lupus Erythematodes, Rheumatoid Arthritis, Celiac Disease and Antiphospholipid Syndrome

**DOI:** 10.3390/vaccines10020333

**Published:** 2022-02-18

**Authors:** Christoph Thurm, Annegret Reinhold, Katrin Borucki, Sascha Kahlfuss, Eugen Feist, Jens Schreiber, Dirk Reinhold, Burkhart Schraven

**Affiliations:** 1Institute of Molecular and Clinical Immunology, Medical Faculty, Otto-von-Guericke-University, 39120 Magdeburg, Germany; christoph.thurm@med.ovgu.de (C.T.); annegret.reinhold@med.ovgu.de (A.R.); sascha.kahlfuss@med.ovgu.de (S.K.); dirk.reinhold@med.ovgu.de (D.R.); 2ChaMP, Center for Health and Medical Prevention, Otto-von-Guericke-University, 39106 Magdeburg, Germany; 3Health Campus Immunology, Infectiology and Inflammation (GC-I3), Medical Faculty, Otto-von-Guericke-University, 39120 Magdeburg, Germany; eugen.feist@med.ovgu.de (E.F.); jens.schreiber@med.ovgu.de (J.S.); 4Institute of Clinical Chemistry and Pathobiochemistry, Medical Faculty, Otto-von-Guericke-University, 39120 Magdeburg, Germany; katrin.borucki@med.ovgu.de; 5Institute of Medical Microbiology and Hospital Hygiene, Medical Faculty, Otto-von-Guericke University, 39120 Magdeburg, Germany; 6Department of Rheumatology, Helios Specialist Hospital Vogelsang, 39245 Gommern, Germany; 7Department of Pneumology, University Hospital, Otto-von-Guericke-University, 39120 Magdeburg, Germany

**Keywords:** SARS-CoV-2, COVID-19, COVID-19 vaccination, autoimmunity, antiphospholipid syndrome, CCP, collagenosis, celiac disease

## Abstract

The COVID-19 pandemics has caused the death of almost six million people worldwide. In order to establish collective immunity, the first vaccines that were approved in Germany were the vector virus-based vaccine Vaxzevria and the mRNA vaccines Comirnaty and Spikevax, respectively. As it was reported that SARS-CoV-2 can trigger autoimmunity, it is of significant interest to investigate whether COVID-19 vaccines evoke the formation of autoantibodies and subsequent autoimmunity. Here, we analyzed immune responses after different vaccination regimens (mRNA/mRNA, Vector/Vector or Vector/mRNA) with respect to anti-SARS-CoV-2-specific immunity and the development of autoantibodies well known for their appearance in distinct autoimmune diseases. We found that anti-SARS-CoV-2 antibody levels were 90% lower after Vector/Vector vaccination compared to the other vaccinations and that Vector/mRNA vaccination was more effective than mRNA/mRNA vaccination in terms of IgM and IgA responses. However, until 4 months after booster vaccination we only detected increases in autoantibodies in participants with already pre-existing autoantibodies whereas vaccinees showing no autoantibody formation before vaccination did not respond with sustained autoantibody production. Taken together, our study suggests that all used COVID-19 vaccines do not significantly foster the appearance of autoantibodies commonly associated with lupus erythematodes, rheumatoid arthritis, Celiac disease and antiphospholipid-syndrome but provide immunity to SARS-CoV-2.

## 1. Introduction

At the end of 2019, the first cases of a novel disease causing flu-like symptoms appeared. The disease evoked acute respiratory distress and cumulated in the death of up to 8% of the infected individuals [1]. Lateron, a novel and highly contagious strain of betacoronaviruses, the severe acute respiratory syndrome coronavirus type 2 (SARS-CoV-2), was isolated and identified as the pathogen responsible for the disease, which was consecutively named coronavirus disease-19 (COVID-19). Since the beginning of the resulting SARS-CoV-2 pandemic, the WHO reported more than 400 million COVID-19 cases and almost 5.8 million deaths as a direct consequence of COVID-19 [2].

Individuals infected with SARS-CoV-2 exhibit a broad spectrum of clinical symptoms. However, in approximately 50% of all cases, the infection shows an inapparent disease course, potentially favoring the undetected and uncontrolled spreading of the virus. Symptomatic SARS-CoV-2 infections are characterized by symptoms such as fever, fatigue, dry cough, headache, myalgia, sore throat, nausea and/or diarrhea [3,4]. In severe cases, COVID-19 is not only limited to the lungs but can also affect the skin, the kidneys, the nervous system and the hematological system [5]. These SARS-CoV-2 infections often show a remarkable imbalance in the immune response, e.g., a hyperactivation of neutrophils and an excessive production of proinflammatory cytokines. Of note, the latter clinical signs and reactions are also frequently detected in patients suffering from autoimmune diseases [6]. Thus, from an immunological point of view, the immune reaction in response to SARS-CoV-2 infections shares similarities to autoimmune diseases. In this context it was reported that COVID-19 patients carrying antinuclear antibodies (ANAs), antiphospholipid antibodies or anti-SS-A/Ro show more severe COVID-19 courses than patients devoid of these autoantibodies [7,8,9,10]. Interestingly, in non-symptomatic COVID-19 patients and patients with only mild disease courses, an age-dependent tendency towards the development of autoantibodies such as anti-cyclic citrullinated peptides (CCP)-IgG and anti-tissue transglutaminase (TTG)-IgA antibodies was also detected [11].

To date, priming of specific immune responses by active immunization represents the standard and most successful prophylactic strategy to stop the pandemic of SARS-CoV-2. Therefore, different COVID-19 vaccines including mRNA vaccines, viral vectors, inactivated SARS-CoV-2 or protein-based vaccines were developed [12]. These efforts resulted in the approval of three particular COVID-19 vaccines by the end of 2020 in Germany. Two of them, BioNTech’s Comirnaty and Moderna’s Spikevax, use the mRNA technique to induce anti-SARS-CoV-2 immunity whereas Vaxzevria developed by AstraZeneca uses a non-replicating chimpanzee adenovirus. All three vaccines appeared safe and showed significant efficacy in clinical trials upon homologous prime/boost vaccinations [13,14,15]. However, it represents a significant gap in our knowledge that systematic clinical studies assessing whether COVID-19 vaccination also triggers the production of autoantibodies are missing yet, although clear correlation of severe COVID-19 disease courses and the appearance of autoantibodies was shown. Clinical studies aiming at elucidating this important question would be of significant interest, especially in view of very recent studies reporting myocarditis after mRNA vaccination especially in younger (<30 years) male individuals [16,17,18,19]. We here analyzed anti-SARS-CoV-2 responses and concomitant production of autoantibodies frequently associated with lupus erythematodes, rheumatoid arthritis, celiac disease and antiphospholipid syndrome in healthcare workers before and after applying heterologous and homologous prime-boost vaccination protocols against COVID-19.

## 2. Materials and Methods

*Participants.* The participants of this manuscript are from the CoVac study that was conducted between April and October 2021 as a prospective, observational study surveying the induction of autoimmunity upon anti-COVID-19 vaccination. Participants were recruited within the employees of the University Hospital Magdeburg and the Medical Faculty of the Otto von Guericke University, Magdeburg, based on prime injections and/or decision for booster injections. Participants received prime injections with Spikevax, Vaxzevria or Comirnaty. Homologous booster doses following primes by Spikevax and Comirnaty were applied after 28 and 41 days, respectively. After Vaxzevria prime, participants could opt for a Spikevax or Vaxzevria booster 80 days after the prime. Participants who received a homologous vaccination with either Spikevax or Comirnaty were combined in the mRNA/mRNA group (Table 1). After informed consent and a questionnaire to assess local and systemic symptoms after vaccination, we collected blood samples by venipuncture followed by serum collection. Participants were 36% male and 64% female with a mean age of 36 years (20–65 years) for the mRNA/mRNA group, 47 years (24–69 years) for the Vector/Vector group and 38 years (19–66 years) for the Vector/mRNA group. 

*Quantification of Anti-SARS-CoV-2-Sp1 antibodies.* Quantification of anti-SARS-CoV-2-Sp1-IgG antibodies was conducted using EliA SARS-CoV-2-Sp1 IgG test (ThermoFisher Scientific, Freiburg, Germany) according to the manufacturer’s instructions. Samples exceeding the range of the test were diluted (1:2, 1:5 or 1:10) using sample diluent (ThermoFisher Scientific, Freiburg, Germany). To allow a better inter-assay comparison of the results, values of the EliA SARS-CoV-2-Sp1 IgG test were transposed from EliA U/mL to the BAU/mL of the WHO International Standard according to the following equation: 1 EliA U/mL = 4 BAU/mL. Quantification of anti-SARS-CoV-2-IgG/IgA/IgM was performed using the Elecsys Anti-SARS-CoV-2 assay (Roche Diagnostics, Rotkreuz, Switzerland). Samples were diluted 1:20 or 1:50. The cutoffs for positive samples were ≥28 BAU/mL for the EliA und ≥0.8 BAU/mL for the Elecsys, respectively.

*Determination of neutralizing antibodies.* The neutralization of the binding of the SARS-CoV-2-spike protein to human ACE2 was analyzed using the SARS-CoV-2-NeutraLISA (Euroimmun, Lübeck, Germany) according to the instruction by the manufacturer. Prior to analysis, serum samples were diluted 1:5 in sample diluent. Samples with a relative neutralization of ≥25% were considered positive.

*ELISA.* The presence of autoantibodies against cyclic citrullinated peptide (CCP, IgG; Medipan, Dahlewitz, Germany) and tissue transglutaminase (TTG, IgA; Generic Assays, Dahlewitz, Germany) were analyzed by ELISA according to the instructions of the manufacturer. Serum samples were diluted 1:100 and analyzed in duplicates. Cut-off for positivity were set at ≥30 U/mL for anti-CCP and ≥20 U/mL for anti-TTG.

Quantifications of autoantibodies against Cardiolipin, Prothrombin and β2-Glycoprotein were conducted by ELISA using the Random Access Analyzer Alegria (Orgentec, Mainz, Germany) according to the manufacturer’s instructions. The following test strips were used: Anti-Cardiolipin Screen (ORG 215S), Anti-beta-2-Glycoprotein I Screen (ORG 221S), Prothrombin screen (ORG 241S). The applied cutoffs were: anti-Cardiolipin-Screen ≥ 10 U/mL, anti-β2-Glycoprotein ≥ 10 U/mL, Prothrombin screen ≥ 20 U/mL.

*IFN-γ release assay.* Cellular immunity upon vaccination against SARS-CoV-2 was analyzed using the Quan-T-cell SARS-CoV-2 system and the Quan-T-cell ELISA (Euroimmun, Lübeck, Germany) according to the manufacturer’s instructions. Samples exceeding the detection limits were diluted 1:2 or 1:5. For the analysis of the data, individual blank values were subtracted from the IFN-γ concentrations upon stimulation with SARS-CoV-2 peptides.

*Immunofluorescence.* Serum samples were screened for the presence of autoantibodies against nuclear antigens (ANA) using immunofluorescence with the ANA HEp-2 plus kit (Generic Assays, Dahlewitz, Germany) according to the manufacturer’s instructions. Briefly, serum samples were initially diluted 1:80 in sample buffer. Positive samples were further diluted until 1:2560. The slides were evaluated using fluorescence microscopy by two independent observers.

*Statistics.* Statistical analysis was performed using Prism8 (GraphPad, San Diego, CA, USA). Normal distribution of data sets was tested with a Shapiro–Wilk test and significance analysis was performed using a Mann–Whitney test, Kruskal–Wallis test or mixed-effects analysis with Tukey’s multiple comparison test.

*Study approval.* This study was approved by the ethics board of the Medical Faculty, Otto von Guericke University, Magdeburg (certificate 67/21) and the Paul Ehrlich Institute (NIS 613). Written informed consent was obtained from all participants before their enrollment in the study. The study was performed in accordance with the guidelines for good clinical practice and the Declaration of Helsinki.

## 3. Results

### 3.1. Characterization of the Study Cohort

Between April and October 2021, healthcare professionals of the University Hospital and the Medical Faculty of the Otto von Guericke University in Magdeburg, Germany, were recruited to participate in the CoVac study. Following prime vaccination, volunteers were assigned to three individual study groups based on anti-COVID-19 booster vaccination. Participants that received a homologous vaccination with Spikevax (*n* = 25) or Comirnaty (*n* = 16) were pooled in the homologous mRNA group (mRNA/mRNA, *n* = 41). The homologous vaccination group with Vaxzevria (Vector/Vector) included 38 participants whereas 42 participants underwent a heterologous prime-boost vaccination regimen by first receiving Vaxzevria followed by Spikevax (Vector/mRNA) (Table 1). One participant from the Vector/Vector group was excluded from the study as a result of a current immunosuppressive therapy. The basic characteristics of the three groups are summarized in Table 1. First serum samples were obtained 33.2 days (±6.9 days), 77.0 days (±9.7 days) and 78.9 days (±5.9 days) after the prime for the mRNA/mRNA, Vector/Vector and Vector/mRNA group, respectively. Timing differences between the groups are a result of the guidelines for timing of booster vaccination following vector or mRNA vaccination provided by the German standing committee on vaccination (STIKO) of the Robert Koch Institute, Berlin [20].

### 3.2. Both Heterologous and Homologous Prime-Boost Immunization Strategies Provide Humoral and Cellular Immunity to SARS-CoV-2

To monitor the anti-SARS-CoV-2 antibody response upon immunization, serum samples were obtained one day prior to the booster as well as 14, 28 and 120 days after the booster (Figure 1A). Samples were analyzed for the presence of anti-SARS-CoV-2-Sp1-IgG antibodies. Importantly, we already detected significant differences between the study groups in the samples that were taken one day before the boost. Indeed, upon vector prime, 31 samples (39%) did not show anti-Sp1-IgG antibodies above the cutoff, while following mRNA prime only one participant did not produce specific IgG antibodies (2%) (Figure 1B). Further, the mean antibody concentration was 10 times lower in the vector group compared to the mRNA group (48.2 BAU/mL, 95%CI 39.4–56.9 and 527.3 BAU/mL, 95%CI 367.0–687.7, respectively).

Homologous and heterologous prime-boost vaccinations induced an increase in the Sp1-IgG antibody levels in all three study groups. However, homologous vector vaccination appeared less potent with regard to antibody production compared to homologous or heterologous prime-boost vaccination using an mRNA vaccine. Indeed, the mean antibody concentration in the Vector/Vector group at 14 days post booster was only 171.6 BAU/mL (95%CI 113.9–229.4) whereas it was 2351 BAU/mL (95%CI 1944–2758) in the mRNA/mRNA and 2462 BAU/mL (95%CI 1876–3047) in the Vector/mRNA group. No difference between the two groups after mRNA prime was detected (Figure 1C). We obtained similar results when we analyzed the pan-anti-SARS-CoV-2-Sp1 antibody response (without discrimination between different isotypes). Here, mean antibody concentrations in the Vector/Vector group were also significantly lower (1872 BAU/mL, 95%CI 1274–2471) compared to all other groups (Vector/mRNA 17,244 BAU/mL, 95%CI 14,930–19,557 and mRNA/mRNA 8370 BAU/mL, 95%CI 6534–10,206) (Figure 1D). Using the latter assay we detected significantly increased antibody production upon heterologous prime-boost immunization compared to homologous mRNA vaccination (Figure 1C,D). Additionally, at 28 days post vaccination the antibody concentrations in the Vector/Vector group were significantly lower compared to the other two groups (Figure 1C). Importantly, we observed significantly increased antibody levels in the Vector/mRNA group compared to the mRNA/mRNA group when measuring anti-SARS-CoV-2-Sp1 antibodies (Figure 1D). Notably, anti-Sp1-IgG antibody levels in the Vector/Vector group (154.4 BAU/mL, 95%CI 103.5–205.4) remained stable between day 14 and 28 post booster whereas they declined in the mRNA/mRNA (1714 BAU/mL, 95%CI 1404–2025) and Vector/mRNA (1640 BAU/mL, 95%CI 1227–2053) groups (Figure 1C).

In order to analyze the long-term immune response after the three different vaccination regimens, we quantified the anti-SARS-CoV-2-Sp1 antibodies four months post booster vaccination. In all three study groups, we detected a significant reduction in the antibody concentrations (Figure 1C,D). As observed directly after booster vaccination, also after four months antibody concentrations with both assays were significantly lower in the Vector/Vector group (63.2 BAU/mL, 95%CI 42.9–83.6 and 662.4 BAU/mL, 95%CI 450.5–874.4, respectively) compared to the mRNA/mRNA (453.6 BAU/mL, 95%CI 359.6–547.7 and 2154 BAU/mL, 95%CI 1817–2490, respectively) or the Vector/mRNA group (349.8 BAU/mL, 95%CI 285.4–414.1 and 3388 BAU/mL, 95%CI 2800–3977, respectively). Interestingly, after four months the Sp1-IgG antibodies in the mRNA/mRNA group had already declined to the pre-boost levels whereas the other two vaccination groups remained above the pre-boost level. We still observed significantly increased pan-Sp1-antibody response in the Vector/mRNA group compared to the mRNA/mRNA. 

Recent studies had shown that complete vaccination (prime and boost) using vector vaccines induces a weaker generation of neutralizing antibodies compared to mRNA vaccines [21,22,23,24,25]. Indeed, in our study, the vector group also exhibited significantly reduced neutralization compared to the mRNA group (21.0% inhibition, 95%CI 17.6–24.3 and 64.1% inhibition, 95%CI 54.4–73.7, respectively) (Figure 1E). Although we observed a significant increase in neutralization at 14 days post boost, regardless of the used vaccine (Figure 1F), virus neutralization was again significantly lower in the Vector/Vector group compared to the mRNA/mRNA group (75.2% inhibition, 95%CI 67.9–82.5 and 99.4% inhibition, 95%CI 99.2–99.6, respectively). Interestingly, with a heterologous prime-boost immunization a level of neutralization equal to the mRNA/mRNA group was achieved (99.5% inhibition, 95%CI 99.4–99.7). At 28 days post boost, neutralizations in the mRNA/mRNA and Vector/mRNA groups (98.95 inhibition, 95%CI 98.3–99.6 and 99.1% inhibition, 95%CI 98.7–99.6, respectively) were still superior compared to the Vector/Vector group (72.0% inhibition, 95%CI 64.7–79.4). The analysis after four months post booster vaccination showed decreased virus neutralization in all three groups compared to four weeks post booster. Still, the mRNA/mRNA and Vector/mRNA groups exhibited comparable levels (89.5% inhibition, 95%CI 84.8–94.1 and 90.2% inhibition, 95%CI 86.5–94.0, respectively) whereas the virus neutralization in the Vector/Vector group had dropped by roughly 50% (37.8% inhibition, 95%CI 26.7–49.0). Importantly, almost 50% of the individuals in the Vector/Vector group exhibited virus neutralization below 25%, a level that is possibly insufficient for protection against the infection.

When analyzing the influence of age and gender on antibody production, we observed a slight decrease in anti Sp1-IgG levels in participants ≥50 years within the mRNA/mRNA group compared to the respective younger subgroup at 28 and 120 days post booster (Appendix A). Despite being only significant at this time point, we found this trend for all time points in the mRNA/mRNA and vector/mRNA groups. In contrast, following vector boost we noted a tendency of elevated anti-Sp1-IgG antibodies in participants ≥50 years compared to younger counterparts (Appendix A). We did not see any changes with respect to gender for the anti-Sp1-IgG or for the pan-anti-Sp1-antibody response as well as for neutralizing antibodies for both gender and age (Figure 1B–F).

To also analyze the cellular immunity upon different vaccination strategies, we quantified the IFN-γ release by T cells upon stimulation with SARS-CoV-2 peptides four months post vaccination. As observed for the antibody response, the cellular immune response was also significantly weaker after Vector/Vector vaccination compared to mRNA/mRNA or Vector/mRNA vaccination (Figure 1G). Indeed, we detected an IFN-γ release in the Vector/Vector group of 330.7 mIU/mL (95%CI 239.8–421.6), whereas the IFN-γ concentration was three times higher after mRNA/mRNA (1057 mIU/mL, 95%CI 722.4–1391) or nine times higher after Vector/mRNA vaccination (2650 mIU/mL, 95%CI 1905–3396). Importantly, four months post booster, cellular immunity upon heterologous prime-boost vaccination was significantly stronger than after mRNA/mRNA vaccination.

### 3.3. Reactogenicity

Reactogenicity was assessed 1 day before (for the prime) and 14 as well as 28 days after boost. After the prime doses, frequencies of local and/or systemic reactions were comparable between participants receiving mRNA or vector vaccination, with a minority experiencing no reactions and the majority experiencing local and systemic reactions (Figure 2A). However, after vector prime more participants reported only systemic reactions (e.g., fever, fatigue, chills, headache, malaise or limb pain) whereas after mRNA prime more local reactions were noted. The severity of symptoms after the prime was higher in the vector group, as the majority of participants in this group (>50%) reported more than three symptoms whereas more than 70% of the vaccinees described less than three symptoms after mRNA prime (Figure 2B). All the reported reactions to the vaccines were mild to moderate and none were severe. Compared to vector vaccination, local reactions were more frequent after prime vaccination with an mRNA vaccine as reflected by increased rates of tenderness/pain and erythema (Figure 2C). In contrast, vector prime vaccination induced more systemic reactions such as fever, fatigue, chills, headache, malaise or limb pain (Figure 2D).

Importantly, we observed strong differences between the three study groups after booster vaccination. Thus, booster reactions after homologous vector/vector vaccination were much milder compared to the reactions after homologous mRNA/mRNA or heterologous prime-boost immunization (Figure 2A). In contrast, the frequency of local and systemic reactions increased after the mRNA boost compared to the prime vaccinations for both groups. In addition, the severity of the reactions in the mRNA/mRNA and Vector/mRNA groups were higher compared to the Vector/Vector group and, in the case of the mRNA/mRNA group, to the prime vaccination. In both groups, >50% of the participants experienced three or more symptoms (Figure 2B). With respect to the frequencies of individual symptoms, no major differences between the mRNA/mRNA and Vector/mRNA groups were detected (Figure 2C,D). In conjunction and as reported before by others [22] our data confirm that prime vaccination with an mRNA vaccine induces milder reactions compared to the vaccination with a vector vaccine. In contrast, after homologous or heterologous booster vaccination with an mRNA vaccine, the severity of reactions increased compared to homologous vaccination with a vector vaccine. However, no major differences between the mRNA/mRNA and Vector/mRNA group were observed.

### 3.4. Homologous and Heterologous Prime-Boost Immunization Strategies Do Not Induce the Generation of Autoantibodies That Typically Accompany Lupus Erythematodes, Rheumatoid Arthritis, Celiac Disease and Antiphospholipid Syndrome

In the course of infections with SARS-CoV-2, but also upon vaccination, the induction of autoantibodies against various autoantigens has been described [6,7,8,26]. However, systematic clinical studies investigating autoantibody production upon vaccination in a larger cohort are missing so far. Therefore, we set out to analyze whether the booster vaccination may trigger an increase in the serum concentration of distinct autoantibodies. First, we studied the profile of anti-Cardiolipin (CARD), anti-Prothrombin (PRO) and anti-β2-Glycoprotein (β2GP) autoantibodies, which are important parameters in the diagnosis of Antiphospholipid syndrome (APS) [27]. Upon booster vaccination, two participants that had been negative after the first vaccination developed autoantibodies above the cutoff. Both individuals belonged to the Vector/Vector group. Among these, one participant showed a transient appearance of CARD autoantibodies at 14 days post booster, which declined below the cutoff at 28 days post booster (Figure 3A). The second participant developed PRO autoantibodies after booster vaccination, which stayed elevated at 14 and 28 days post booster (Figure 3B). Moreover, individuals with preexisting CARD antibodies showed a tendency of increased autoantibody concentrations throughout the observation period (Figure 3A and Table 2). This could indicate that the stimulation of the immune system by the booster vaccination with mRNA or vector vaccines also drives the concomitant production of pre-existing CARD autoantibodies.

Next, we evaluated the changes in autoantibody concentrations of the whole cohort. We did not observe significant increases for CARD and PRO autoantibodies after the mRNA/mRNA or Vector/Vector vaccinations (Figure 3A,B). However, in the Vector/mRNA group the levels of CARD and PRO autoantibodies decreased four months post vaccination compared to 28 days post vaccination or before booster and 14 days post booster, respectively. Detailed analyses with respect to the age of the individuals revealed that the changes in CARD autoantibodies predominantly occur in individuals <50 years (Appendix A). In the Vector/mRNA group, we detected increased PRO autoantibodies in female participants compared to the respective male cohort (Appendix A). The analysis of β2GP autoantibodies in the Vector/mRNA group revealed a slight but significant increase at 14 days post booster (Figure 3C). This effect was not age dependent as individuals below and above 50 years showed the same tendency. Interestingly, this trend was exclusively observed among the female participants (Appendix A). Individuals with preexisting β2GP autoantibodies did not show increasing levels within the observation period, in contrast to the CARD autoantibodies (Figure 3C and Table 2).

The analysis of the autoantibody profile in non-symptomatic and COVID-19 patients with a mild course of the disease had shown age-dependent elevations of anti-CCP-IgG and anti-TTG-IgA autoantibodies [11]. Therefore, we analyzed if the three different vaccination regimens would trigger similar events. However, we did not observe a relevant induction in anti-CCP-IgG or anti-TTG-IgA antibodies irrespective of the vaccination regime, age or gender (Figure 3C,D and Appendix A). In the mRNA/mRNA (CCP and TTG-IgA) and in the Vector/mRNA group (TTG-IgA), autoantibody levels were slightly increased 120 days post vaccination compared to the other time points but were still below the cutoff. Booster vaccination led to a slight increase in anti-TTG-IgA levels in one volunteer with pre-existing autoantibodies in the Vector/mRNA group that were not associated with any clinical signs.

The analysis of antinuclear antibodies (ANA) is an excellent tool to detect a variety of autoantibodies. We therefore determined ANAs in the different study groups. We detected ANAs in the serum of 20 participants at least at one time point. Of note, the ANA-positive participants were equally distributed within the three study groups (eight participants from the mRNA/mRNA group, nine participants from the Vector/Vector group and eight participants from the Vector/mRNA group, respectively). Figure 3F summarizes titer values and respective ANA patterns of all tested samples. Amongst these, we detected an increase in ANA titers two and four weeks after booster vaccination in only five participants. Three of these belonged to the mRNA/mRNA group and showed ANA positivity at only one time point with a maximal titer of 1:160 and a speckled pattern. In one participant within the Vector/Vector group, the titer increased from 1:160 before booster to 1:320 at time points 14 and 28 days post booster and declined to 1:160 at the time point 120 days post booster. In these samples, we detected nuclear dots, which were not specific to the Sp100 antigen. Notably, in the same samples we also detected an increase in PRO autoantibodies. In the Vector/mRNA group, one participant exhibited a centromere-like pattern with increasing titers starting at 1:160 before the booster and reaching 1:320 post booster, which decreased to 1:160 (speckled) 120 days post booster (Figure 3F) and, furthermore, in five volunteers (2× mRNA/mRNA and 3× Vector/Vector group) that were negative before we detected ANAs 120 days post booster. These samples exhibited cytosolic or speckled patterns with a maximal titer of 1:320.

Table 2 summarizes the results of the autoantibody analyses for nine participants showing an increase in at least one of the tested autoantibodies upon booster vaccination. Four of these belonged to the mRNA/mRNA group. However, three participants only showed a slight induction of ANA titers up to 1:160. In a recent study, 22.5% of the tested healthy controls exhibited comparable titer values [11]. Therefore, the increases in ANA titers detected in these three individuals were possibly not triggered by the booster vaccination. When these three individuals are excluded from the evaluation, it becomes evident that all but one participant showing an increase in autoantibodies were already positive before the booster vaccination.

## 4. Discussion

The appearance of SARS-CoV-2 and the COVID-19 pandemic fueled the development of vaccines with the aim of establishing specific anti-COVID-19 immunity as the first line of defense. To this end, novel vaccination techniques such as mRNA- or vector-based vaccines were developed and some of them have shown a high efficiency and safety profile [12]. By December 2020, three different vaccines had been approved for vaccination against COVID-19 in Germany. Two of these were mRNA vaccines (Comirnaty and Spikevax, respectively) while the third one was a vector-based vaccine (Vaxzevria). All three vaccines confer great protection against COVID-19. However, due to the novelty of these vaccines little is known about important questions such as the dynamics of virus-specific antibody responses or a potential contribution to the deregulation of the immune system. In order to shed some light on this unexplored terrain, we systematically analyzed antibody responses after different prime/boost regimens with respect to the intended protective immunity against COVID-19 vs. the unintended induction of autoimmunity, whereby we intentionally focused on some of the most frequent autoimmune diseases (lupus erythematodes, rheumatoid arthritis, celiac disease and antiphospholipid syndrome).

The question of antibody production upon anti COVID-19 vaccination arose early after the first vaccination campaigns had started because serum antibody concentrations are widely used as correlates of protection for various other viral infections such as measles, HPV, pertussis or smallpox [28,29,30,31]. Our analyses revealed a remarkable difference in the production of SARS-CoV-2-specific IgG antibodies when comparing homologous vaccination with either mRNA or vector-based vaccines. Here, the amount of antibodies produced in the Vector/Vector group reflected only 10% of that seen in the mRNA group and, further, virus neutralization appeared significantly less effective. These observations confirm previous studies [21,22,23,24,25]. Interestingly, in our study, the heterologous prime-boost immunization strategy was as effective in the induction of anti-Sp1-IgG antibodies as a homologous mRNA vaccination (Figure 1C) despite the fact that the antibody level before boost as well as the interval between the prime and the boost and were different. These observations are in line with other reports [21,24,25]. However, using a second assay that does not exhibit a specificity for any immunoglobulin isotype we detected higher antibody levels upon heterologous prime-boost immunization (Figure 1D). These differences can be attributed either to more pronounced IgA and IgM responses upon heterologous prime-boost vaccination or to differences in the spectrum of antibodies induced by the vaccination regimens leading to altered sensitivity of the used tests. However, samples from both groups showed similar inhibition in the surrogate neutralization test indicating that the differences in the pan-Sp1-antibody response are not relevant for virus neutralization. Additionally, our analyses also revealed that T-cell-mediated immune responses are more pronounced after heterologous Vector/mRNA prime-boost vaccination. In conjunction, our data indicate that heterologous prime-boost immunization regimens might represent an improved strategy for the stimulation of the immune system and hence lead to superior long-term protection against COVID-19. This information might be of importance especially for the ongoing second booster vaccinations. It is likely that also a heterologous second booster vaccination after a homologous prime/boost regimen yields in an improved humoral and cellular immune response. To clarify this question further, studies after second booster vaccinations are required.

Longitudinal analyses of anti-SARS-CoV-2 immune responses following different vaccination regimens are of significant importance due to the fact that stopping the COVID-19 pandemic will most likely require long-term immunity, especially against severe forms of the disease. To address this question, we set out and quantified the anti-SARS-CoV-2-Sp1 antibodies four months post vaccination. Our data show that in all study groups, antibody values had already significantly declined at this time point but, importantly, were still above the cut-off (especially if the volunteers had received at least one vaccination with an mRNA vaccine). Our observation is further in line with reports that emerged from other studies [32,33,34,35,36]. It is important to consider that a durable immune response is likely also triggered when a primary infection is followed by a vaccination booster [32,36,37]. Indeed, in some cases, this sequence of events appears to be beneficial for the establishment of immunological memory. However, the quality of immune priming by a SARS-CoV-2 infection likely also depends on the severity of the infection, which makes the long-term immunity after this kind of immunization difficult to predict, unlike the situation that occurs after a prime-boost vaccination regimen.

In the context of SARS-CoV-2 infections, various reports described the appearance of autoantibodies such as ANA, anti-phospholipid antibodies or SS-A. Furthermore, the appearance of these autoantibodies correlated with a more severe course of the disease [7,8,9]. In order to identify the molecular and cellular basis underlying these observations, mechanisms that are known to induce autoimmunity upon viral infections, e.g., molecular mimicry, have been assessed for COVID-19. Many hexa- and heptapeptides originating from human chaperones, olfactory receptors or different membrane proteins are also part of the SARS-CoV-2 proteome [38,39,40,41]. However, the biological relevance of these structures has not been investigated so far. Additionally, many case reports and small studies have tried to establish a connection between COVID-19 and autoimmunity [42,43,44,45]. However, none of the currently existing studies clearly showed that COVID-19 indeed augments autoimmune diseases, and systematic studies investigating autoantibodies following different vaccination regimens are currently elusive. It is important to note that a coincidence of events does not necessarily imply causality, which means that in some cases COVID-19 and autoimmunity could represent completely unrelated events that by chance appear in parallel. Alternatively, pre-existing subclinical autoimmunity also might contribute to these observations. Nevertheless, the discussion about a possible interconnection between autoimmunity and COVID-19 triggered the fear that vaccination might induce autoimmunity [46]. This is further emphasized by recent reports about rare cases of Myocarditis in young male individuals upon vaccinations with mRNA vaccines [16,17,18,19]. To our knowledge, our study is the first that systematically addresses this issue. Importantly, we did not detect an induction of autoantibodies that normally accompany the autoimmune diseases lupus erythematodes, rheumatoid arthritis, celiac disease and antiphospholipid syndrome upon booster vaccination in all three study groups. Only participants with pre-existing autoantibodies responded with detectable increases in autoantibody production upon vaccination, a phenomenon that has been described for other vaccinations before [47]. However, one participant with a positive ANA result before booster vaccination developed anti-Prothrombin antibodies thereafter, which stayed elevated at least until 28 days post booster but returned to below the cutoff four months post vaccination. Nevertheless, none of the participants had a history of autoimmunity or developed clinical signs that would indicate the occurrence of an autoimmune disease after the booster. Therefore, it is questionable whether the autoantibodies that we detected during our study are of medical relevance. Furthermore, APS antibodies can also be induced by various triggers, for example viral infections [48]. Therefore, it cannot be ruled out that vaccination-unrelated factors/events might be responsible for the observed changes in autoantibody levels. Still, it is important to note that the induced autoantibodies either declined to baseline levels or showed no further increase during the observation period or that this increase was not associated with clinical signs. This might indicate that the appearance of these particular autoantibodies is a transient event that occasionally accompanies the vaccination but does not have clinical relevance. However, we certainly cannot exclude the possibility that autoantibodies that have not been analyzed in this study are induced upon vaccination or that more severe and possibly clinically relevant autoimmunity is triggered after repetitive vaccinations with the same vaccines or following vaccination with new vaccines that are developed in the future with the aim to fight novel COVID-19-mutants. We are aware that the explanatory power of this study is limited due to the relatively small group sizes and the short observation period. Hence, to clarify the question whether COVID-19 vaccination can cause autoimmunity, future analyses of greater cohorts appear to be mandatory. These studies should also include a healthy control group and predisposed individuals in order to improve the predictive power. 

## Figures and Tables

**Figure 1 vaccines-10-00333-f001:**
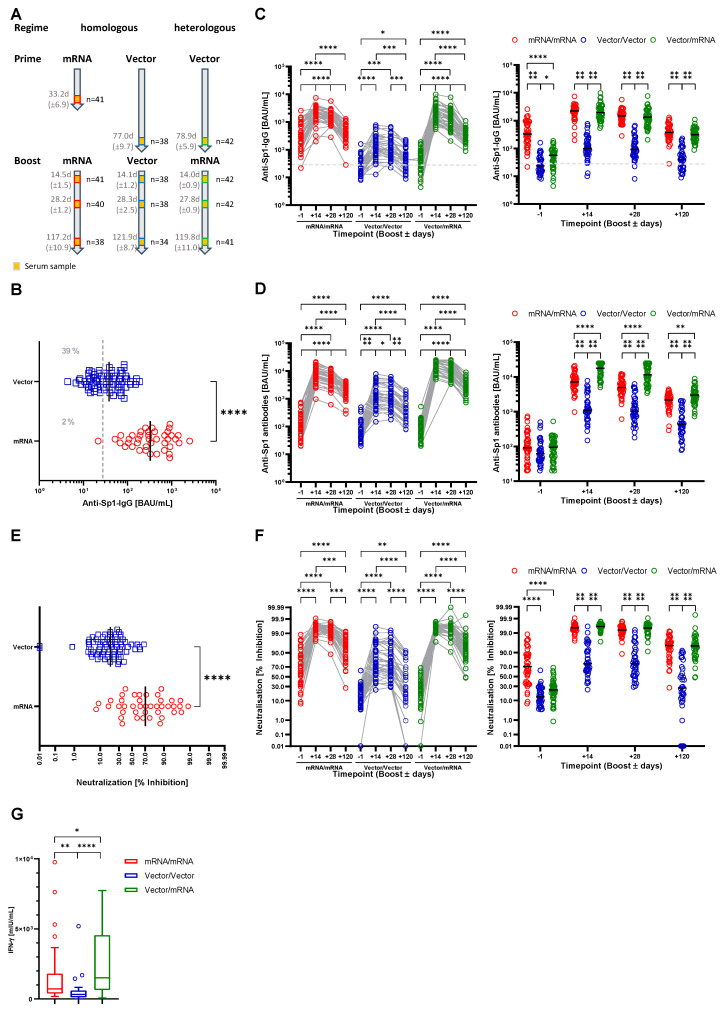
Homologous and heterologous prime-boost immunization regimens induce anti-SARS-CoV-2 antibody production and immunity. (**A**) Study design and sample generation. Based on the received vaccinations, participants were grouped into the mRNA/mRNA, Vector/Vector or Vector/mRNA group. Serum samples were obtained one day before as well as 14, 28 and 120 days after booster. (**B**) Serum levels of anti-SARS-CoV-2-Sp1-IgG antibodies after vector (blue) or mRNA (red) prime one day before booster vaccination. The dashed line indicates the applied cutoff for positivity. (**C**) Left: Profile of serum anti-SARS-CoV-2-Sp1-IgG antibodies of individual participants based on the different vaccination strategies (red-mRNA/mRNA; blue-Vector/Vector; green-mRNA/Vector). Connected circles represent antibody levels of individual participants. Right: Serum anti-SARS-CoV-2-Sp1-IgG antibodies according to the study groups at the different time points. (**D**) Left: Profile of serum anti-SARS-CoV-2-Sp1 antibodies of individual participants based on the different vaccination strategies and indicated groups. Connected circles represent antibody levels for individual participants. Right: Serum anti-SARS-CoV-2-Sp1 antibodies according to the study groups at the different time points. (**E**) Levels of neutralizing antibodies after Vector (blue) or mRNA (red) prime one day before booster vaccination. (**F**) Left: Profile of neutralizing antibodies of individual participants within the indicated groups. Connected circles represent the antibody levels of an individual participant. Right: Neutralizing antibodies according to the study groups at the different time points. (**F**) IFN-γ release upon stimulation with SARS-CoV-2 peptides four months post booster vaccination (red-mRNA/mRNA, blue-Vector/Vector, green-mRNA/Vector). Outliers have been identified by ROUT method and excluded from statistical analysis. Statistical analyses: Mann–Whitney test (**B**,**E**), Mixed-effects analysis with Tukey’s multiple comparison test within and between groups (**C**,**D**,**F**) and Kruskal–Wallis test (**G**). * *p* < 0.05, ** *p* < 0.01, *** *p* < 0.001, **** *p* < 0.0001.

**Figure 2 vaccines-10-00333-f002:**
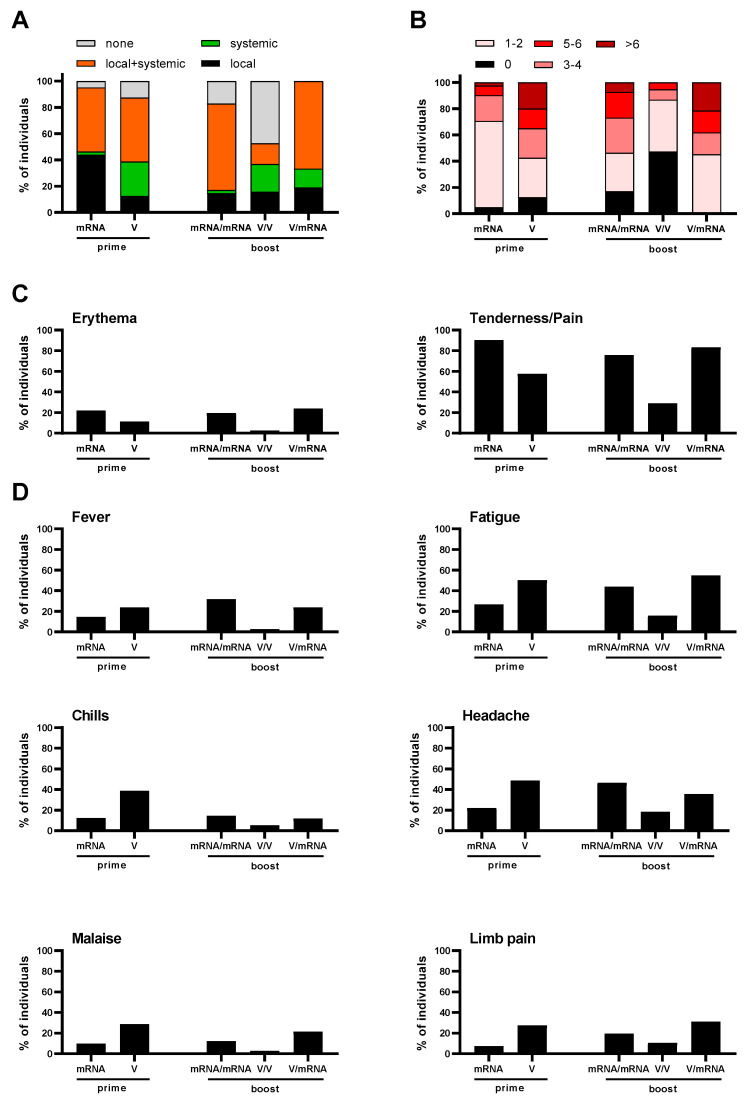
Reactogenicity after prime vaccinations with an mRNA or Vector vaccine and after booster vaccination for the mRNA/mRNA, Vector/Vector (V/V) and Vector/mRNA (V/mRNA) groups. (**A**) Analysis of local and systemic reactions; (**B**) analysis of the severity of reactions based on the number of symptoms; (**C**) analysis of the frequencies of local reactions; (**D**) analysis of the frequencies of systemic reactions.

**Figure 3 vaccines-10-00333-f003:**
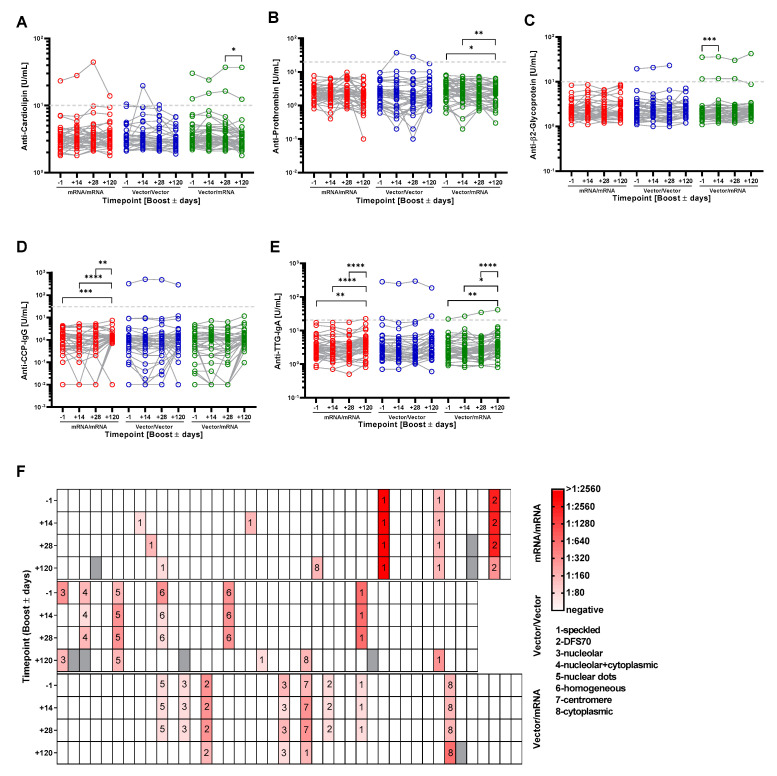
Analysis of autoantibody levels in the serum of the participants at the different time points and in the different groups (red-mRNA/mRNA; blue-Vector/Vector; green-Vector/mRNA). Connected circles represent the antibody levels of an individual participant. (**A**) anti-Cardiolipin; (**B**) anti-Prothrombin; (**C**) anti-β2-Glycoprotein; (**D**) anti-CCP; (**E**) anti-TTG autoantibody levels in the indicated groups. (**F**) Analysis of ANAs in serum samples of the shown groups. Color of the boxes indicate individual titers and numbers state the respective patterns. Statistical analyses by Mixed-effects analysis with Tukey’s multiple comparison test within and between groups (**C**). * *p* < 0.05, ** *p* < 0.01, *** *p* < 0.001, **** *p* < 0.0001.

**Table 1 vaccines-10-00333-t001:** Basic characteristics of the study groups. Spx—Spikevax, Com—Comirnaty.

	mRNA/mRNA *n* = 41		Vector/Vector	Vector/mRNA
Subgroup	Spx/Spx *n* = 25	Com/Com *n* = 16	*n* = 38	*n* = 42
Age (years)	35.9 (±13.3)		47.2 (±13.5)	37.9 (±14.1)
Sex				
Male	16 (39%)		15 (39%)	13 (31%)
Female	25 (61%)		23 (61%)	29 (69%)
Boost (Prime + xx days)	34.2 (±6.8)		78.0 (±10.0)	80.3 (±5.8)
1st serum sample (Prime + xx days)	33.2 (±6.9)		77.0 (±9.7)	78.9 (±5.9)
2nd serum sample (Boost + xx days)	14.5 (±1.5)		14.1 (±1.2)	14.0 (±0.9)
3rd serum sample (Boost + xx days)	28.2 (±1.2)		28.3 (±2.5)	27.8 (±0.9)
4th serum sample (Boost + xx days)	117.2 (±10.9)		121.9 (±8.7)	119.8 (±11.0)
Boost − 1 days	*n* = 41		*n* = 38	*n* = 42
Boost + 14 days	*n* = 41		*n* = 38	*n* = 42
Boost + 28 days	*n* = 40		*n* = 38	*n* = 42
Boost + 120 days	*n* = 38		*n* = 34	*n* = 41

**Table 2 vaccines-10-00333-t002:** Summary of participants with increasing values for selected autoantibodies. Bold values are above the respective cutoff. Sp—speckled; Dots—nuclear dots; cen—centromere; ho—homogeneous.

Participant No.	Study Group	Autoantibody 1			Autoantibody 2		
		Autoantibody	Time Point	Result	Autoantibody	Time Point	Result
6	mRNA/mRNA	Cardiolipin	−1	**23. U/mL**			
			+14	**28 U/mL**			
			+28	**44.3 U/mL**			
			+120	**13.9 U/mL**			
9	mRNA/mRNA	ANA	−1	negative			
			+14	**1:80 sp**			
			+28	negative			
			+120	negative			
10	mRNA/mRNA	ANA	−1	negative			
			+14	negative			
			+28	**1:160 sp**			
			+120	negative			
20	mRNA/mRNA	ANA	−1	negative			
			+14	**1:160 sp**			
			+28	negative			
			+120	negative			
39	Vector/mRNA	Cardiolipin	−1	**30.3 U/mL**	β2-Glycoprotein	−1	**35.2 U/mL**
			+14	**24 U/mL**		+14	**36 U/mL**
			+28	**37.2 U/mL**		+28	**29.9 U/mL**
			+120	**37 U/mL**		+120	**42.3 U/mL**
52	Vector/Vector	Prothrombin	−1	9.4 U/mL	ANA	−1	**1:160 Dots**
			+14	**37.4 U/mL**		+14	**1:320 Dots**
			+28	**28.3 U/mL**		+28	**1:320 Dots**
			+120	17.7 U/mL		+120	**1:160 Dots**
70	Vector/mRNA	TTG-IgA	−1	**21.9 U/mL**			
			+14	**26.7 U/mL**			
			+28	**34.6 U/mL**			
			+120	**41.2 U/mL**			
81	Vector/Vector	CCP	−1	**325 U/mL**	ANA	−1	**1:320 ho**
			+14	**510 U/mL**		+14	**1:320 ho**
			+28	**482 U/mL**		+28	**1:320 ho**
			+120	**289 U/mL**		+120	negative
86	Vector/Vector	Cardiolipin	−1	2.8 U/mL			
			+14	**19.7 U/mL**			
			+28	6.9 U/mL			
			+120	3.4 U/mL			

## Data Availability

All data for this study are contained within the manuscript and the Appendix A.

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
