# Peer review of "Homologous and Heterologous Anti-COVID-19 Vaccination Does Not Induce New-Onset Formation of Autoantibodies Typically Accompanying Lupus Erythematodes, Rheumatoid Arthritis, Celiac Disease and Antiphospholipid Syndrome"

_vaccines, 2022, doi:10.3390/vaccines10020333_

Round 1

Reviewer 1 Report

The work by Thurm et al. was well conducted and sounds clear. It shows  that all used combinations of SARS-CoV2 vaccines besides providing immunity to SARS-CoV 2 do not induce autoantibody production. The later is public health matter and an important information that contribute to dissipate concerns on side effects of large scale vaccination. 

Author Response

We thank the reviewer for this positive evaluation of our work.

Reviewer 2 Report

Thank you for the opportunity of reviewing this paper. The research is interesting but there are a number of points that require revision.

Please, find here my suggestions/comments.

First of all, anti-SARS-CoV-2 vaccine (e.g., titles and lines 26, 36, and throughout the whole paper) must be changed to COVID-19 vaccine. Indeed, the protection offered by the current developed and authorized vaccine is only against the consequences of the disease, not against infection sustained by SARS-CoV-2 (as observed with the breakthrough infections). This is a very relevant difference that must be addressed before considering the paper for publication.

Similarly, (line 56) COVID-19 infection should be corrected to COVID-19 cases, SARS-CoV-2 infections, COVID-19…

Despite being commonly entered the scientific community, the term heterologous vaccination is ontologically wrong, referring to the use of a microorganism to confer protection against another (e.g., use of cowpox virus to immunize against smallpox infection). In case of COVID-19 vaccination it should be considered as mix-and-match or, as scientific papers are reporting, heterologous prime-boost immunization.

Line 48, please update epidemiological data.

Please, compare your results about vaccine response with important evidence in the field; for example, let me suggest the from a comprehensive ongoing longitudinal study (VASCO project) on the response to BNT162b2 COVID-19 vaccine (of which I am NOT an author):

- Ponticelli D, Madotto F, Conti S, Antonazzo IC, Vitale A, Della Ragione G, Romano ML, Borrelli M, Schiavone B, Polosa R, Ferrara P, Mantovani LG.

Response to BNT162b2 mRNA COVID-19 vaccine among healthcare workers in Italy: a 3-month follow-up. Intern Emerg Med. 2021 Oct 12:1-6. doi: 10.1007/s11739-021-02857-y

- Ponticelli D, Antonazzo IC, Caci G, Vitale A, Della Ragione G, Romano ML, Borrelli M, Schiavone B, Polosa R, Ferrara P. Dynamics of antibody response to BNT162b2 mRNA COVID-19 vaccine after 6 months. J Travel Med. 2021;28(8):taab173. doi: 10.1093/jtm/taab173.

Please, better discuss the role of previous infection as immune priming (the suggested references are both useful also for this point).

Line 357. “We suggest that individuals older than 60 years of age represent the most important target group for intensification of vaccination” should be rephrased to “Our findings suggest that COVID-19 vaccination is an effective tool for the protection against the consequences of the disease, allowing to describe the most vulnerable groups that should be offered with vaccines, including individuals aged 60 years or older and people with chronic conditions.

Author Response

Thank you for the opportunity of reviewing this paper. The research is interesting but there are a number of points that require revision.

Please, find here my suggestions/comments.

First of all, anti-SARS-CoV-2 vaccine (e.g., titles and lines 26, 36, and throughout the whole paper) must be changed to COVID-19 vaccine. Indeed, the protection offered by the current developed and authorized vaccine is only against the consequences of the disease, not against infection sustained by SARS-CoV-2 (as observed with the breakthrough infections). This is a very relevant difference that must be addressed before considering the paper for publication.

Similarly, (line 56) COVID-19 infection should be corrected to COVID-19 cases, SARS-CoV-2 infections, COVID-19…

Despite being commonly entered the scientific community, the term heterologous vaccination is ontologically wrong, referring to the use of a microorganism to confer protection against another (e.g., use of cowpox virus to immunize against smallpox infection). In case of COVID-19 vaccination it should be considered as mix-and-match or, as scientific papers are reporting, heterologous prime-boost immunization.

  • We thank the reviewer for these important points and apologize for the imprecise wording. We have changed the respective phrases throughout the manuscript accordingly.

Line 48, please update epidemiological data.

  • We have updated the data (as of February 14th, 2022)

Please, compare your results about vaccine response with important evidence in the field; for example, let me suggest the from a comprehensive ongoing longitudinal study (VASCO project) on the response to BNT162b2 COVID-19 vaccine (of which I am NOT an author):

- Ponticelli D, Madotto F, Conti S, Antonazzo IC, Vitale A, Della Ragione G, Romano ML, Borrelli M, Schiavone B, Polosa R, Ferrara P, Mantovani LG.

Response to BNT162b2 mRNA COVID-19 vaccine among healthcare workers in Italy: a 3-month follow-up. Intern Emerg Med. 2021 Oct 12:1-6. doi: 10.1007/s11739-021-02857-y

- Ponticelli D, Antonazzo IC, Caci G, Vitale A, Della Ragione G, Romano ML, Borrelli M, Schiavone B, Polosa R, Ferrara P. Dynamics of antibody response to BNT162b2 mRNA COVID-19 vaccine after 6 months. J Travel Med. 2021;28(8):taab173. doi: 10.1093/jtm/taab173

Please, better discuss the role of previous infection as immune priming (the suggested references are both useful also for this point).

  • We have added a paragraph comparing the results of our study with other longitudinal studies focusing on humoral immunity and the impact of SARS-CoV-2 infections prior vaccination.

Line 357. “We suggest that individuals older than 60 years of age represent the most important target group for intensification of vaccination” should be rephrased to “Our findings suggest that COVID-19 vaccination is an effective tool for the protection against the consequences of the disease, allowing to describe the most vulnerable groups that should be offered with vaccines, including individuals aged 60 years or older and people with chronic conditions.

  • We agree with the reviewer that COVID-19 vaccination is an effective tool for the protection of several risk groups and not only for individuals older than 60 years. In the current manuscript we, however, have not claimed that vaccination should be only considered for individuals older than 60 years of age. More precisely, the sentence “We suggest that individuals older than 60 years of age represent the most important target group for intensification of vaccination” cited by the reviewer is not part of our manuscript. Thus, we did not address it.

Reviewer 3 Report

Well built studio with a high amount of data. The results of the antibody response, both humoral and cellular, as a consequence of the various types of vaccination are very well exposed. However, the data relating to the production of autoantibodies are less convincing, and above all the assertion, already expressed in the title of the article, that vaccination does not induce the production of autoantibodies. A more nuanced approach, both in the conclusions and in the title, would be more consistent with the data reported.

Author Response

We thank the reviewer for this valuable opinion. We agree that the presentation of the data about autoantibodies needed some rephrasing in order to point out our findings more clearly. Therefore, we have changed the title of the manuscript and also rephrased respective passages in the text so that the studied autoantibodies and the associated diseases are more pronounced. By doing so, we now avoid global statements about autoimmunity in the context of COVID-19 vaccinations, for which our study does not provide enough evidence. We thank the reviewer for pointing this out.